# Genome mining of cyclodipeptide synthases unravels unusual tRNA-dependent diketopiperazine-terpene biosynthetic machinery

Tingting Yao[1], Jing Liu[1], Zengzhi Liu[1], Tong Li[1], Huayue Li[1,2], Qian Che[1,2], Tianjiao Zhu[1,2], Dehai Li [1,2], Qianqun Gu[1,2] & Wenli Li [1,2]

Cyclodipeptide synthases (CDPSs) can catalyze the formation of two successive peptide bonds by hijacking aminoacyl-tRNAs from the ribosomal machinery resulting in diketopiperazines (DKPs). Here, three CDPS-containing loci (*dmt1–3*) are discovered by genome mining and comparative genome analysis of *Streptomyces* strains. Among them, CDPS DmtB1, encoded by the gene of *dmt1* locus, can synthesize *cyclo*(L-Trp-L-Xaa) (with Xaa being Val, Pro, Leu, Ile, or Ala). Systematic mutagenesis experiments demonstrate the importance of the residues constituting substrate-binding pocket P1 for the incorporation of the second aa-tRNA in DmtB1. Characterization of *dmt1–3* unravels that CDPS-dependent machinery is involved in CDPS-synthesized DKP formation followed by tailoring steps of prenylation and cyclization to afford terpenylated DKP compounds drimentines. A phytoene-synthase-like family prenyltransferase (DmtC1) and a membrane terpene cyclase (DmtA1) are required for drimentines biosynthesis. These results set the foundation for further increasing the natural diversity of complex DKP derivatives.

[1] Key Laboratory of Marine Drugs, Ministry of Education of China, School of Medicine and Pharmacy, Ocean University of China, 266003 Qingdao, China. [2] Laboratory for Marine Drugs and Bioproducts of Qingdao National Laboratory for Marine Science and Technology, 266003 Qingdao, China. These authors contributed equally: Tingting Yao, Jing Liu. Correspondence and requests for materials should be addressed to W.L. (email: liwenli@ouc.edu.cn)

Natural products featuring 2,5-diketopiperazine (DKP) scaffolds are structurally diverse and widely produced by bacteria, fungi, plants, and animals[1]. The core DKP scaffold makes these privileged structures resistant to proteolysis and capable of crossing the intestinal barrier and blood–brain barrier (BBB)[2]. Various compounds based on naturally occurring DKPs have recently gained much attention and are currently being investigated as potential anticancer drugs, antibiotics, and anti-inflammatory agents, such as plinabulin (NPI-2358)[3], phenylahistin[4], and bicyclomycin[5].

The DKP-ring system is formed through double condensation of two α-amino acids via basically two mechanisms: non-ribosomal peptide synthetase (NRPS)[6] and cyclodipeptide synthase (CDPS)[7,8]. Unlike NRPSs, which use free amino acids by activating them through adenylation, CDPSs hijack aminoacyl-tRNAs (aa-tRNAs) from their primary use in the translation process[9]. Functionally characterized CDPSs are essentially from three bacteria phyla (Actinobacteria, Firmicutes, and Proteobacteria)[10,11]. Albeit displaying low sequence identity (mostly less than 30%), they fall into two main phylogenetically distinct subfamilies, namely NYH and XYP[10–12]. CDPSs characterized to date are able to generate different DKPs, consisting of 18 of the 20 proteinogenic amino acids[10,11] (with the exception of Lys and Asp) and 26 noncanonical amino acids[13]. The crystallographic structures, as well as mutagenesis studies revealed that the two aa-tRNAs bind at different substrate-binding sites: pocket P1 and pocket P2. The specificity of the first aa-tRNA depends on its aminoacyl moiety, conversely that of the second aa-tRNA lies on both the aminoacyl moiety and its tRNA sequence[14]. Based on multiple sequence alignments of functionally characterized CDPSs, Jacques et al. proposed the sequence signatures of P1 and P2 associated with the formation of DKP product[11]. However, the impact of mutagenesis of the residues in P2 along with the interaction between P1 and P2 is still obscure.

Attachment of various side chains, such as methyl, prenyl, and hydroxyl, enables DKPs to interact with an array of biological targets[1]. As a matter of fact, CDPSs generally occur associated with tailoring enzymes[15]. Nonetheless only four CDPS-dependent pathways have been fully understood so far, including albonoursin[7,16], nocazines[17,18], pulcherriminic acid[19], and bicyclomycin[20,21]. Compared to the canonic NRPS pathways, CDPS pathways are still underexplored. The type of CDPS-synthesized DKPs combined with modifications would greatly expand the chemical diversity of DKP-based compounds.

In this study, genome mining of marine *Streptomyces* strains reveals a three-gene CDPS-containing locus (named *dmt1*) in *Streptomyces youssoufiensis* OUC6819 (formerly named *Streptomyces* sp. CHQ-64[22]). In silico analysis of the *Streptomyces* genomes in GenBank uncovers another two CDPS-containing loci homologous to *dmt1* from *Streptomyces* sp. NRRL F-5123 (named *dmt2*) and *Streptomyces aidingensis* CGMCC 4.5739 (named *dmt3*). Phylogenetic analysis reveals that these three CDPSs (named DmtB1–3) belong to the cWX-forming group of the NYH subfamily. Functional characterization demonstrates that DmtB1 is a CDPS synthesizing *cyclo*(L-Trp-L-Xaa) (X = Val, Pro, Leu, Ile, or Ala) with cWV being the major product. The underlying molecular basis for substrate specificity of DmtB1 is then probed via extensive mutagenesis of the amino acid residues in the two binding pockets. Furthermore, these three CDPS-dependent biosynthetic loci are characterized and revealed a CDPS-dependent pathway whereby the DKP product is prenylated and cyclized by tailoring enzymes to afford terpenylated diketopiperazine compounds, dubbed drimentines (DMTs, Fig. 1). Notably, a phytoene-synthase-like (PSL) family

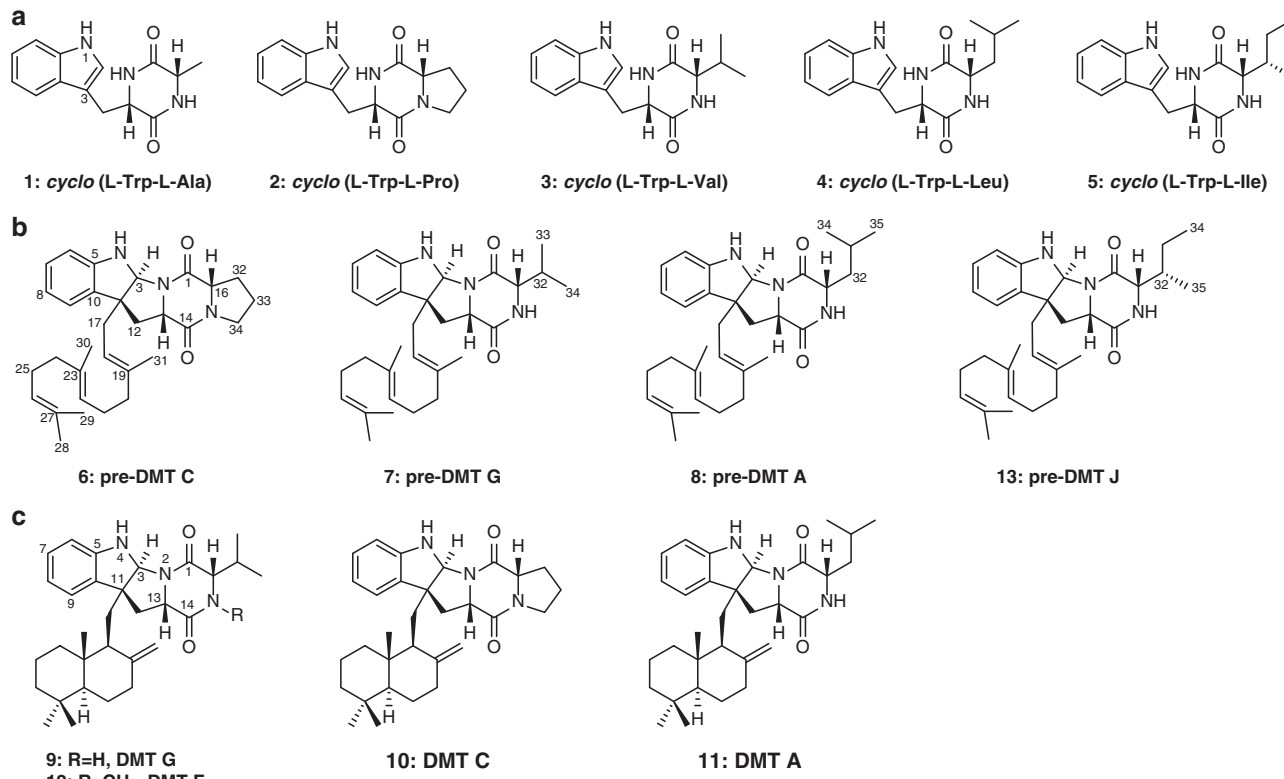

**Fig. 1** Chemical structures of compounds **1–13**. **a** Diketopiperazines (DKPs) synthesized by DmtB1 (**1–5**); **b** pre-drimentines (pre-DMTs, **6–8** and **13**); **c** drimentines (DMTs, **9–12**)

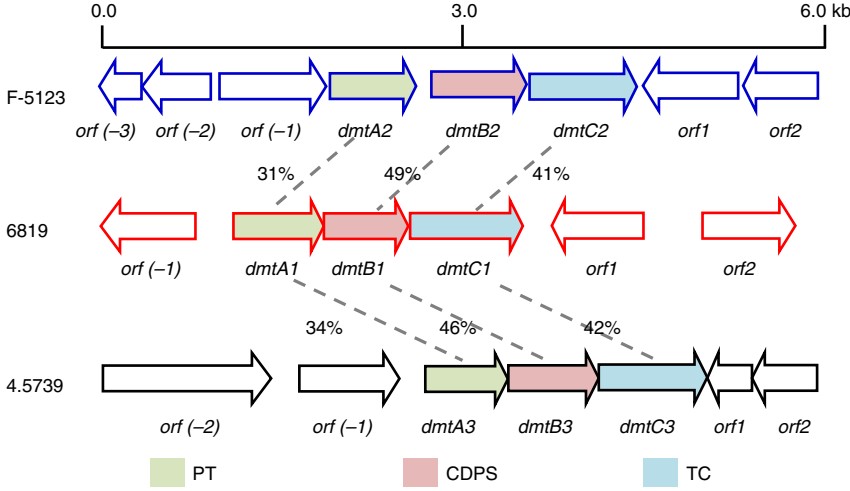

**Fig. 2** Comparison of the three CDPS-containing loci *dmt1–3*. The *dmt1–3* are from *S. youssoufiensis* OUC6819, *Streptomyces* sp. NRRL F-5123 and *S. aidingensis* CGMCC 4.5739, respectively

| Table 1 Predicted functions of the *dmtA1-C1* | | | | |
|---|---|---|---|---|
| **Protein** | **Size (aa)** | **Proposed function** | **Homologs** | |
| | | | **Protein/organism** | **Accession no. (Identity/similarity %)** |
| DmtA1 | 250 | Hypothetical protein | XiaE/*Streptomyces* sp. HKI0576 | CCH63731.1 (30/42) |
| DmtB1 | 233 | Cyclodipeptide synthase, CDPS | Amir_4627/*Actinosynnema mirum* DSM 43827 | ACU38460.1 (37/52) |
| DmtC1 | 311 | Phytoene synthase | NzsG/*Streptomyces* sp.MA37 | ALL53320.1 (21/33) |

prenyltransferase (PT) and a membrane terpene cyclase (TC) are identified. Thus, this study expands our understanding of CDPSs, as well as CDPS-dependent biosynthetic pathways, setting the stage to further increase the natural diversity of complex DKP derivatives using combinatorial approaches.

## Results

**Genome mining reveals three homologous CDPS-containing loci.** Genome mining of marine-derived *S. youssoufiensis* OUC6819 revealed a putative three-gene operon *dmt1*, containing a CDPS gene *dmtB1* (Fig. 2). An additional two homologous loci were then identified in *Streptomyces* sp. NRRL F-5123 (*dmt2*) and *S. aidingensis* CGMCC 4.5739 (*dmt3*) following in silico analysis of *Streptomyces* genomes available on GenBank (Fig. 2). DmtB1 displayed 37% identity to Amir_4627 (ACU38460.1) (Table 1), 49% to DmtB2, and 46% to DmtB3 (Fig. 2). Phylogenetic analysis revealed that DmtB1–3 belong to the NYH subfamily and are clustered with cWX-synthesizing CDPSs (Supplementary Fig. 1). Notably, DmtB1 bears a Gln203 (numbering after AlbC[14]) instead of His203, which is strictly conserved among the NYH subfamily[11]. To investigate their encoding DKP product(s), the *dmtB1–3* genes were cloned into pET-28a and subsequently introduced into *E. coli* BL21(DE3). High-performance liquid chromatography (HPLC) analysis of the culture supernatants showed that five DKP compounds **1–5** were produced by *E. coli* cells containing *dmtB1* (Fig. 3). These compounds were identified as *cyclo*(L-Trp-L-Ala) (**1**), *cyclo*(L-Trp-L-Pro) (**2**), *cyclo*(L-Trp-L-Val) (**3**), *cyclo*(L-Trp-L-Leu) (**4**), and *cyclo*(L-Trp-L-Ile) (**5**) based on their retention times (Fig. 3) and their high-resolution mass spectra (HR-MS, Supplementary Figs. 3–7) in comparison to those of authentic standards. Conversely, expression of *dmtB2*

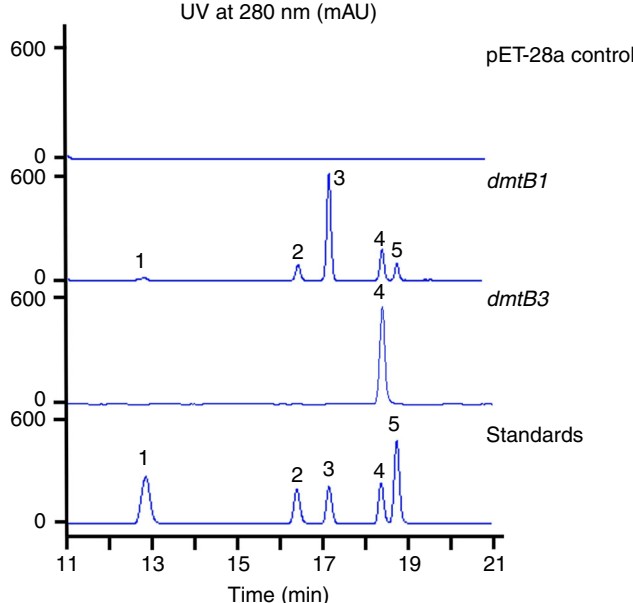

**Fig. 3** HPLC traces of culture supernatants of *E. coli* cells expressing DmtBs

yielded **2** plus a minor amount of **4** (Supplementary Fig. 2), consistent with the recently published result[23], and DmtB3 only synthesized **4** (Fig. 3). These results highlighted the substrate promiscuity of DmtB1 regarding to the second amino acid residue incorporated into DKPs. In contrast, DmtB2 and DmtB3 exhibit a relatively stringent specificity.

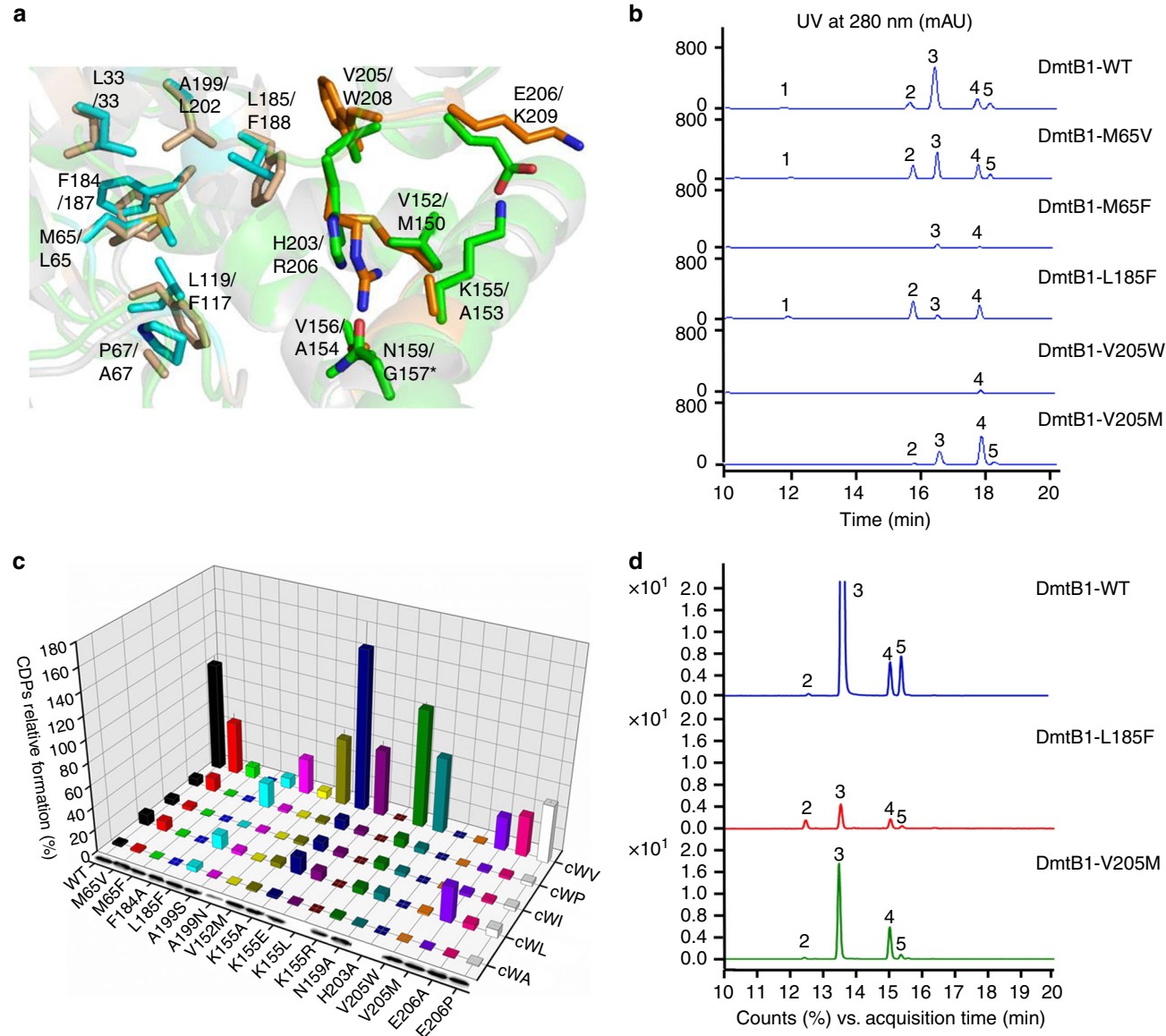

**Fig. 4** Mutagenesis of the residues located in the binding pockets of DmtB1. **a** Superposition of the binding pockets in the DmtB1 model with those in the crystal structure of YvmC. The P1 and P2 of the DmtB1 model are shown in cyan and green, respectively; the P1 and P2 of YvmC (PDB id: 3OQJ) are shown in wheat and brown, respectively. **b** HPLC traces of the products formation by DmtB1 and its variants (also see Supplementary Fig. 10). **c** Histogram of the relative formation of various DKPs synthesized by DmtB1 and its variants after 20 hrs of expression in *E. coli* at 16 °C. The corresponding western blots indicating amounts of proteins are also shown. Error bars represent ± s.d. of three independent experiments. **d** Total ion chromatogram in multiple reaction monitoring (MRM) mode for compounds **2**–**5** resulting from in vitro assays of DmtB1 and its variants (L185F and V205M). The relative titers of compounds were compared based on the peak area of MS$^2$ fragment 130.11 from each molecular ion

**Probing the molecular basis for DmtB1 substrate specificity**. With the three orthologous CDPSs in hand, we probed the underlying molecular basis for the substrate specificity. Firstly, the binding pockets P1 and P2, accommodating the aminoacyl moieties of the first and second aa-tRNAs, were predicted by multiple sequence alignment (Supplementary Fig. 8, Supplementary Data 2). As shown in Supplementary Data 2, although P1 and P2 are independent, the residues constituting the P1 pockets of DmtB1–3 are not well conserved with those of the cWW-synthesizing CDPSs. DmtB1–3 harbor a L186 (AlbC numbering) instead of Phe, and contain a hydrophobic residue at position 200 (Ala or Leu) other than hydrophilic residues present in cWW-synthesizing CDPSs. The residues constituting the P2 pockets of DmtB1–3 display a striking difference to those of

cWW-synthesizing CDPSs at position 207 by replacing Pro with an acidic amino acid Glu (in DmtB1 and DmtB3) or Asp (in DmtB2). Except V156, other residues in P2 are dissimilar among the aligned CDPSs.

A structural model of DmtB1 was then generated using the protein threading and fold recognition server I-TASSER[24] to assist understanding the underlying molecular basis of substrate specificity (Supplementary Fig. 9). As shown in Fig. 4a, the binding pockets P1 and P2 of the structural model of DmtB1 superimposed well with those of YvmC[25] (PDB id: 3OQJ). To test the key residue(s) for substrate specificity, we performed a series of mutagenesis of selected residues in both pocket P1 (M65, F184, L185 and A199) and pocket P2 (V152, K155, N159, H203, V205, and E206). The resulting variants were

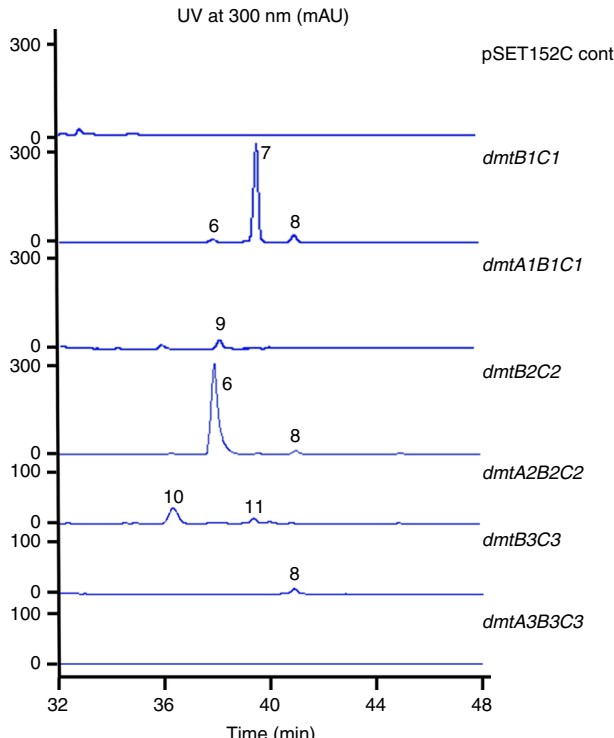

**Fig. 5** Heterologous expression of the *dmt1–3* loci in *S. coelicolor* M1146

introduced into *E. coli*, and in vivo synthesis of DKPs, as well as recombinant protein expression levels were compared to those of the wild-type DmtB1 (Fig. 4, Supplementary Fig. 10).

For the residues in the pocket P1, the variant M65V synthesized increased amount of *cyclo*(L-Trp-L-Pro) (**2**, by ~1.7-fold) but decreased amount of *cyclo*(L-Trp-L-Val) (**3**, by ~2-fold) and *cyclo*(L-Trp-L-Ile) (**5**, by ~2-fold), and conversely, the variant M65F was poorly active despite being abundantly produced (Fig. 4b, c). Substitution of L185 with Phe led to significant change of the production of these five DKPs: inability to synthesize **5**, strong defect to form **3** (decreased by ~10-fold), and yield improvement of **1** and **2** by ~2.4 and 3.1-fold, respectively, and thus the major product switched from **3** to **2**. When V205 in the pocket P2 was mutated into Trp or Met, the variant V205W only synthesized a very low amount of **4** and the major product of the variant V205M shifted from **3** to **4** with increased amount by about 2.3-fold and simultaneously decreased **2**, **3**, and **5** or abolished **1** production (Fig. 4b, c). These results clearly indicate that substitution of residues constituting either P1 or P2 pocket could affect the nature of the second amino acid incorporated.

Considering the in vivo results are dependent on cellular availability of specific aa-tRNA substrates, we further performed in vitro assays for DmtB1 and its variants, L185F and V205M, to closely control the concentrations of the aa-tRNA substrates. As shown in Fig. 4d, in comparison to the wild-type DmtB1, the **3**-forming ability of both variants was significantly decreased by ~43-fold (L185F) and 6-fold (V205M), but **3** still served as the most abundant product; conversely, the **2**-forming ability of L185F was improved by 2.5-fold, and no obvious change was observed for the **4**-forming ability of V205M. Although the major product synthesized by L185F and V205M in vitro is different from that detected in vivo (Fig. 4b), the changes in the formation ratios of **2** vs. **3** and **4** vs. **3** between the wild-type DmtB1 and the variants were consistent in vivo and in vitro (Supplementary

Table 7). Thus, the importance of L185 and V205 for substrate specificity could be safely established.

Furthermore, based on the mutagenesis results for DmtB1, the equivalent residues in DmtB2 (V74 and L194 in pocket P1, and I214 in pocket P2, Supplementary Data 2) were mutated, and the result demonstrated L194 (corresponding to L185 in DmtB1) might serve as an important substrate specificity determinant in DmtB2 in vivo (Supplementary Fig. 11).

**Inspection of the genetic contexts of *dmtB1–3*.** As CDPSs generally occur associated with tailoring enzymes, we next inspected the genetic contexts of the *dmtB1–3*, disclosing the presence of conserved membrane protein DmtA1–3 and phytoene synthase DmtC1–3 (Fig. 2). DmtA1 displayed 30% identity to the membrane terpene cyclase XiaE (CCH63731.1) (Table 1) and showed identities of 31% and 34% to DmtA2 and DmtA3, respectively (Fig. 2). DmtC1 displayed 21% identity to PSL prenyltransferase NzsG (ALL53320.1) (Table 1), and showed identities of 41% and 42% to DmtC2 and DmtC3, respectively (Fig. 2). Hence the *dmt1–3* loci were predicted to encode DKPs with a terpene moiety. Further inspection into the genetic neighborhood of the *dmt1–3* loci showed: (i) genes surrounding the *dmt1* locus displayed no homology to those of the *dmt2/3* loci; (ii) homologs of the two subunits of cyclic dipeptide oxidase (CDO) were found neighboring to the *dmt2/3* loci, suggesting potential α,β-dehydrogenation(s) would happen on the DKP moiety; (iii) the adjacent regions of the *dmt2/3* loci also harbor putative methyltransferase gene(s), sharing low homology to each other and to characterized methyltransferases from CDPS-containing loci[18,26] (Supplementary Fig. 12).

To identify and characterize the metabolic products of these three assemble lines, a series of expression plasmids were constructed. *dmtBC* and *dmtABC* from each cluster were cloned under the control of the *Streptomyces* constitutive promoter P*gapdh*, resulting in pWLI615-620. These constructs were then introduced into the superhost *Streptomyces coelicolor* M1146. As shown in Fig. 5, expression of *dmtB1C1* led to accumulation of three compounds **6–8**, which presumably derive from compounds **2–4**. By further introducing *dmtA1*, a single compound **9** was produced in a low titer, which was identified as drimentine G (DMT G, Fig. 1, Supplementary Figs. 13–15) harboring a cWV derivative moiety and was previously isolated from the fermentation broths of *S. youssoufiensis* OUC6819[22].

For the expression of the *dmt2* locus, M1146/pWLI617 (expressing *dmtB2C2*) produced compounds **6** and **8**, and *dmtA2B2C2* accumulated compounds **10** and **11** (Fig. 5), which were identified to be DMT C and DMT A, respectively, based on comparison with retention times of authentic standards (Supplementary Fig. 13), as well as HR-MS and ¹H NMR data (Supplementary Figs. 16–19). When we expressed *dmtB3C3*, only a small amount of compound **8** was produced, and no product was observed after further introduction of *dmtA3* (Fig. 5). Taken together these results imply that DmtC1 catalyses the condensation of a farnesyl diphosphate (FPP) moiety to *cyclo*(L-Trp-L-Val) (**3**). The resulting compound **7** is then proposed to be cyclized by DmtA1 to yield DMT G (**9**). Similarly, *cyclo*(L-Trp-L-Pro) (**2**) and *cyclo*(L-Trp-L-Leu) (**4**) assembled by DmtB2 are converted into compounds **6** and **8** by DmtC2, before being cyclized by DmtA2 to yield DMT C (**10**) and DMT A (**11**).

**In vivo characterization of the *dmt1* locus.** To establish the function of the *dmt1* locus in the native micro-organism, gene inactivation was then carried out and resulted in the following three mutant strains *S. youssoufiensis* OUC6819 Δ*dmtA1*, Δ*dmtB1* and Δ*dmtC1* (Supplementary Figs. 20–22). HPLC analysis of the

culture extracts from these mutants revealed that production of DMT G (**9**) and DMT F (**12**), both naturally produced by the wild-type strain, was abolished in every mutant strain (Fig. 6). These results demonstrated that DmtA1, DmtB1 and DmtC1 are all essential for DMT biosynthesis in *S. youssoufiensis* OUC6819. Simultaneously, accumulation of compounds **6–8** was observed in the Δ*dmtA1* culture extract (Fig. 6) with an additional compound **13** in comparison to those in *S. coelicolor* M1146 expressing *dmtB1C1* (Fig. 5). The ratios among **6–8** were analogous in both strains, with **7** being the most abundant intermediate. Genetic complementation of the three mutants by overexpression of *dmtA1, dmtB1,* and *dmtC1* were also performed. The results showed that compounds **9** and **12** were overproduced when *dmtA1, dmtB1,* or *dmtC1* were overexpressed in the wild-type strain background (Supplementary Figs. 23–25). The Δ*dmtB1* (Supplementary Fig. 24) and Δ*dmtC1* (Supplementary Fig. 25) mutants complemented with *dmtB1* and *dmtC1*, respectively, were able to produce DMTs but in limited amount. Conversely, Δ*dmtA1* strain complemented with *dmtA1* accumulated both **9** and **12** in much higher yield than those in the wild-type strain. This complemented strain still produced compounds **6–8, 13** as Δ*dmtA1* did (Supplementary Fig. 23). These results further established the essential role of *dmt1* in DMT biosynthesis.

Subsequently, large scale fermentation of the Δ*dmtA1* mutant strain followed by chemical isolation was performed, leading to identification of compounds **6–8, 13**. HR-ESIMS analysis indicated the molecular formula of **6** as $C_{31}H_{41}N_3O_2$ (*m/z* 488.3272 [M + H]$^+$, calcd 488.3277, Supplementary Fig. 27), **7** as $C_{31}H_{43}N_3O_2$ (*m/z* 490.3425 [M + H]$^+$, calcd 490.3434, Supplementary Fig. 34), **8** as $C_{32}H_{45}N_3O_2$ (*m/z* 504.3580 [M + H]$^+$, calcd 504.3590, Supplementary Fig. 41), and **13** as $C_{32}H_{45}N_3O_2$ (*m/z* 504.3588 [M + H]$^+$, calcd 504.3590, Supplementary Fig. 48), respectively. The structures of these compounds were determined by 1D and 2D NMR spectroscopic analysis (Supplementary Table 9, Supplementary Figs. 28–33, 35–40, 42–47, and 49–54). The HMBC correlations from H-3 to C-13 of the DKP ring in each compound suggested the formation of the fused five-membered ring between the indoline and the DKP ring, which is identical to that in DMTs (Fig. 1). In each $^{13}C$ spectrum of **6–8**,

**13**, we observed fifteen carbon signals, including six olefinic carbons, five methylene carbons and four methyl carbons, which were fully connected by COSY (H-17/H-18, H-20/H-21/H-22, and H-24/H-25/H-26) and HMBC correlations, forming a linear farnesyl chain. The HMBC correlations from H-17 to C-3, C-10, C-11, and C-12 undoubtedly revealed that the farnesyl chain was attached to the cyclized-Trp residue in each compound. The main structural differences among **6–8, 13** were attributed to the other amino acid residues comprising the DKP skeletons with Trp. According to the $^{1}H$ spin system of amino acid side chain in COSY, the residues were finally established to be Pro, Val, Leu and Ile for **6–8, 13**, respectively. The absolute configurations of C-3 and C-11 were determined as 3*S* and 11*S*, respectively, by comparison of their NMR values with those of DMT G (**1**)[22]. Since these compounds are encoded by CDPS-dependent pathway, all the amino acid residues were determined to be L-configuration. The configurations of the three double bonds in the farnesyl group were confirmed to be *trans* by NOE correlations of H-17/H-31, H-22/H-24, and H-26/H-28. Thus, **6–8, 13** were finally identified as DMT analogs with linear farnesyl chains, which are proposed the intermediates in DMT biosynthesis. As the potential cyclization product of **13** would be a putative compound DMT J, **6–8, 13** were named pre-drimentine C, G, A, and J, respectively, among which **8** has previously been generated by organic synthesis[27].

**DmtC1 is a prenyltransferase belonging to the PSL family**. Bioinformatic analysis indicated that DmtC1 has 27–38% similarity to phytoene/squalene synthases, which catalyze the conversion of geranylgeranyl pyrophosphate (GGPP) or two FPP to yield phytoene and squalene, respectively. The heterologous expression experiments, as well as *dmtC1* gene inactivation in *S. youssoufiensis* OUC6819 revealed that DmtC1 is undoubtedly involved in the assembly of the FPP group onto the DKP moieties to generate metabolites that act as substrates for the last enzyme in the pathway, DmtA1. Phylogenetic analysis suggested that DmtC1 is clustered with NzsG[28] and squalene/phytoene synthases instead of typical aromatic or indole PTs (Supplementary Fig. 55). To demonstrate the exact function of DmtC1, we overexpressed *dmtC1* in *E. coli* (Supplementary Fig. 56) and carried out biochemical assays. As the Δ*dmtA1* mutant strain accumulated four pre-DMT compounds **6–8** and **13**, all the four corresponding DKP compounds **2–5** were tested as substrates for DmtC1, in the presence of FPP. After incubation with DmtC1 for 1 h at 30 °C, reactions were analyzed by HPLC. As shown in Fig. 7, DmtC1 successfully condensed FPP onto the C-3 position of **2–5** to provide **6–8** and **13**, respectively. The formation of the fused five-membered ring between the indoline and DKP ring (Fig. 7) is possibly a non-enzymatic process or catalyzed by DmtC1 as well. Notably, the relaxed substrate specificity displayed by DmtC1 would enable potential generation of an array of structurally diverse terpenylated Trp-containing DKPs compounds.

**DmtA1 initiates cyclization by protonating C–C double bond**. Heterologous expression and work in *S. youssoufiensis* OUC6819 also clearly demonstrated the function of DmtA1 as a terpene cyclase. Blastp analysis of DmtA1 revealed it is a membrane protein with seven transmembrane helices (Fig. 8, Supplementary Fig. 57) but did not give any hint on its catalytic activity. The function of the closest homologues XiaE[29]/XiaH[30] could not be predicted either. In DMT biosynthesis, cyclization is proposed to be triggered by protonation of the C-C double bond (between C26 and C27) while protonation of an epoxide ring is involved in xiamycin biosynthesis[29,30]. As initial protonation of an isoprene

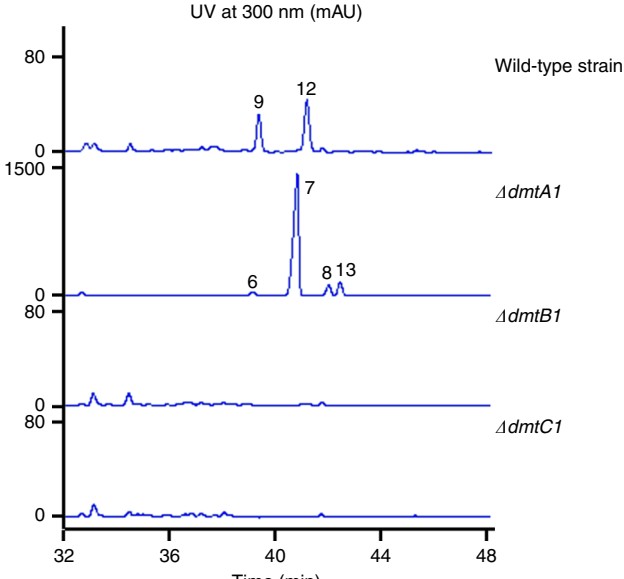

**Fig. 6** HPLC traces of the fermentation products from *S. youssoufiensis* OUC6819 strains

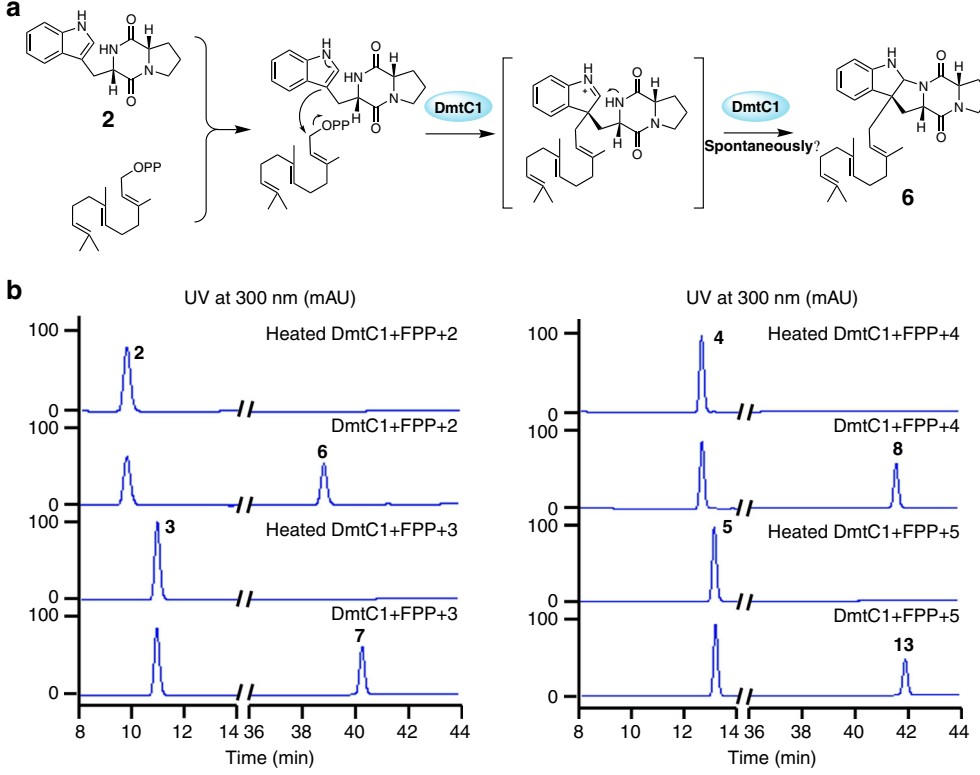

**Fig. 7** Biochemical function of DmtC1. **a** Schematic representation of DmtC1-catalyzed reactions illustrated by using **2** as substrate. **b** HPLC traces of DmtC1 biochemical assays

double bond is less favored than that of an oxirane moiety, we further probed this proposed mechanism for DmtA1-catalyzed cyclization. Sequence alignment with DmtA1–3 (227–250 aa) and XiaE/XiaH, all of which lack the typical aspartate-rich motifs present in TCs, were performed (Supplementary Fig. 58). In addition to the acidic residues E60 and D214 (DmtA1 numbering), conserved in membrane TCs, such as in Pyr4[31] where these residues are proposed to catalyze the cyclization reaction, D94 and E133 were found to be uniquely conserved within DmtA1-3 (Supplementary Fig. 58). These acidic residues were therefore mutated to investigate their potential role in catalysis. Unfortunately, *dmtA1* was unable to express in *E. coli* despite many attempts. As an alternative approach, different versions of mutated *dmtA1* were constructed and introduced into the *S. youssoufiensis* OUC6819 *ΔdmtA1* mutant where the impact of the amino acid substitutions were determined by assessing the catalytic activity of DmtA1 variants. As shown in Fig. 8, substitution of E60 with Ala or Gln completely abolished genetic complementation of the *ΔdmtA1* mutant, undoubtedly supporting its catalytic role in DmtA1. A very low amount of DMT G (**9**) could be detected when using the variant D214A to complement the *ΔdmtA1* strain. Production of **9** was more significantly restored in the strain complemented with the variant D214N. When D94 was mutated, the variant D94A could not complement the *ΔdmtA1* strain while the variant D94N could only form a very limited amount of **9**, suggesting its critical role in catalysis. Substitution of E133 with Ala also led to defective production of **9** as compared to the complemented strain with the wild-type DmtA1.

Other conserved residues were mutated in DmtA1, including W29, Y33, N56, W59 and Y217. As indicated in Fig. 8c and Supplementary Fig. 59, substitution of W29 and W59 with alanine failed to complement the *ΔdmtA1* strain; no production of **9** was observed, supporting their potential role in stabilizing the favorable conformation required for catalysis. Both Y33A and Y33F resulted in inactive DmtA1, indicating the essential role of its hydroxyl group for cyclization, possibly through positioning and increasing the nucleophilicity of the acidic residues. Conversely, Y217A was poorly active and Y217F displayed improved activity but still much lower than that of the wild-type DmtA1. A previously reported the acidity of Asp could be increased by a nearby Cys[32]. The C58 residue unique to DmtA1 (Supplementary Fig. 58) and close to E60 was also mutated to Ala but had almost no influence on its activity. Interestingly, substitution of N56, which is also adjacent to E60, with Ala led to poorly active variant (Fig. 8c, Supplementary Fig. 59), suggesting that might increase E60 acidity rather than C58.

## Discussion

With the increasing availability of genomic information, members of the CDPS family have expanded significantly. Our study revealed two NYH subfamily CDPSs DmtB1 and DmtB3. DmtB1 is a cWX-synthesizing CDPS with cWV being the main major product; while DmtB3 only synthesizes cWL, DmtB2 produces cWP plus a small amount of cWL. Concerning the different substrate specificity exhibited by DmtB1–3, systematic mutagenesis experiments were carried out to establish which key residues impact on specificity. Thus, the important contribution of residues lining the substrate binding pockets P1 and P2 to the incorporation of the second aa-tRNA was established.

Although the first aa-tRNA of DmtB1–3 is the same as that of the other cWW-synthesizing CDPSs, residues lining the binding pocket P1 were not well conserved, indicating that in addition to the pocket P2, the recognition of the second aa-tRNA might not only depend on the residues in P2 but could also be dependent on key residues related to the pocket P1. Variants mutated in P1 such as DmtB1-V65M, DmtB1-L185F (Fig. 4), as well as DmtB2-

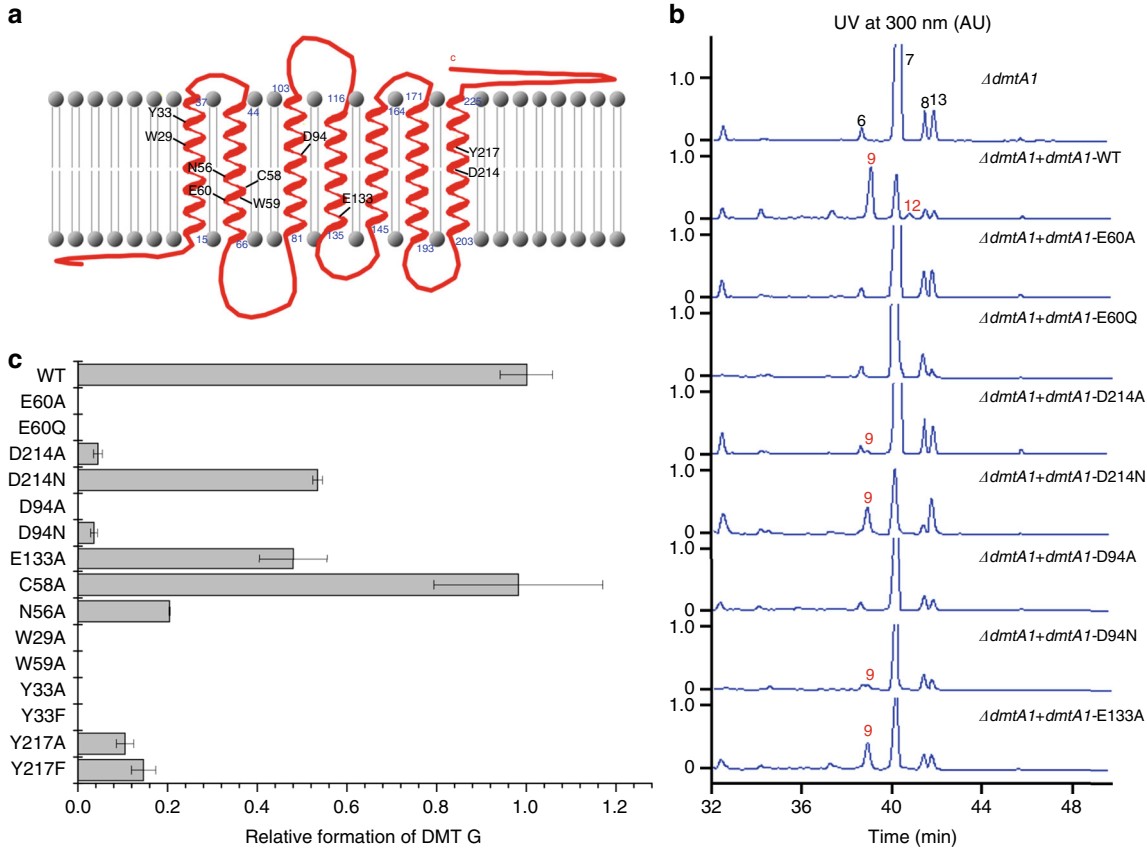

**Fig. 8** Impacts of mutagenised DmtA1 on the final cyclization reaction. **a** Predicted DmtA1 model generated by utilizing TMRPres2D. **b** HPLC traces of Δ*dmtA1* complemented with different *dmtA1* variants (also see Supplementary Fig. 59). The cyclization products **9** and **12** were indicated in red. **c** Relative formation of DMT G (**9**) production in Δ*dmtA1* complemented with different *dmtA1* variants in comparison to complementation with wild-type *dmtA1* (WT). Error bars represent ± s.d. of three independent experiments

L194F (Supplementary Fig. 11) firmly supported this assumption as they led to the production of alternative cWX products. One plausible interpretation is that mutated residues cause an alteration of the binding orientation of the first aa-tRNA substrate and thus directly effect the condensation reactions with the second aa-tRNA. As the binding pocket P2 is formed after the binding of the first aa-tRNA to the pocket P1, changes in specificity regarding the second aa-tRNA might alternatively be mediated by a modification of the binding pocket P2. Future crystallographic structure studies might shed light on these underlying molecular mechanisms. It is worth to mention, products formation by CDPSs in vivo is closely related to the relative abundances of specific aa-tRNAs intracellular[12,14]. The in vivo results showed the production of **3** was significantly impaired in DmtB1-L185F and DmtB1-V205M, and thus the major product changed (Fig. 4b); conversely, the in vitro assays indicated both variants still catalyze formation of **3** the most but in significantly decreased amount (Fig. 4d). The discrepancy between the in vivo and the in vitro data was also observed for AlbC[14]. The changes in the ratios of products formation were thus compared (Supplementary Table 7), which clearly showed the consistency between the in vivo and the in vitro data. Taken together, the importance of L185 and V205 for substrate specificity of DmtB1 was unambiguously established.

CDPS genes are often clustered with other tailoring enzymes to generate complex DKP-containing natural products. Modification of the heterocycle or modification of amino acid side chains of the constituent amino acids can therefore be expected. However, so

far only four CDPS-dependent assembly lines have been fully elucidated, with modifications such as oxidation[16,18,20], and methylation[18]. In this study, we were able to demonstrate that *dmt1* encodes CDPS-dependent machinery involved in DMT biosynthesis both in vivo and in vitro. DmtB1 produces *cyclo*(L-Trp-L-Xaa) (Xaa = Val, Pro, Leu, Ile, and Ala, with cWV being the major product), which undergoes farnesyl transfer followed by cyclization (Fig. 9). Transfer of the farnesyl group is accomplished by a PSL type PT rather than a bacterial indole PT, such as those involved in cyclomarin/cyclomarazine and 6-dimethylallylindole (DMAI)-3-carbaldehyde biosynthesis, CymD[33] and IptA[34], respectively. The only PSL type PT reported so far is NzsG involved in neocarazostatin biosynthesis[28]. NzsG was reported to be highly specific for both the prenyl donor, as well as the prenyl acceptor, transferring dimethylallyl pyrophosphate (DMAPP) onto precarazostatin[28]. In contrast, DmtC1 displays broad substrate specificity regarding to the prenyl acceptor both in vivo (Figs. 5, 6) and in vitro (Fig. 7). Our ongoing study reveals that it also shows promiscuity to the prenyl donor. Therefore, DmtC1 could prove to be a powerful biocatalytic tool for accessing structurally diverse compounds.

Noticeably, both the heterologous host and the native producer only produce DMT compound(s) at relatively low titer (Figs. 5, 6). In contrast, pre-DMTs were able to be accumulated in larger amounts (Figs. 5, 6), which indicates that DmtA1 might be rate limiting during the assembly process. Consistently, the yields of DMTs were significantly improved in the Δ*dmtA1* complemented strain (Supplementary Fig. 23). However, in the *dmtA1* overexpression strain (in

**Fig. 9** The CDPS-dependent assembly line of DMT G, **9**

the wild-type *S. youssoufiensis* OUC6819 background), the yields of DMTs were only slightly improved, which might be due to DmtA1 being a membrane protein. As cyclization of pre-DMTs should be initiated by protonation of the C–C double bond (between C-26 and C-27), which is less favored than protonation of the epoxide ring as that happens during xiamycins biosynthesis[29,30], additional acidic residues may be necessary for catalysis. Our systematic site-directed mutagenesis results demonstrated that, in addition to the conserved E60, the acidic residue D94, unique to DmtA1-3 (absent in XiaE/XiaH), probably plays an important role in catalysis, too (Fig. 8, Supplementary Fig. 58). Very recently, a similar fungal membrane TC MacJ, involved in macrophorins biosynthesis, was identified from *Penicillium terrestris*, which also bears an additional acidic residue Asp96 related to catalysis[35]. The residue Y33 was also found to be conserved among all the membrane TCs and might contribute to increase the nucleophilicity of the acidic residues. Indeed, substitution of Y33 with either Ala or Phe resulted in inactive DmtA1 variants.

While the wild-type *S. youssoufiensis* OUC6819 is also able to produce DMT F (**12**) (Fig. 6, Supplementary Fig. 26), which harbors a methyl group at *N*-15 position (Fig. 1), no methyltransferase gene was found adjacent to the *dmt1* locus (Supplementary Fig. 12), indicating this gene is probably located in another genomic region. The timing for methylation has still to be investigated. The methyltransferase genes located in the adjacent regions of the *dmt2/3* loci, as well as the characterized DKP-ring-modifying *N*-methyltransferase Amir_4628[26] would enlighten deciphering this methylation step. In addition, the presence of the CDO homologs surrounding the *dmt2/3* loci suggest introduction of α,β-dehydrogenation on DMTs, thus generating more complex natural products. Significantly, our studies on *dmt1–3* have expanded our understanding towards CDPS, as well as CDPS-dependent biosynthetic machineries, opening up new possibilities for combinatorial biosynthesis to enrich the structural diversity of complex DKPs compounds.

## Methods

**Materials and general experimental procedures**. Bacterial strains and plasmids used and constructed during this study are listed in Supplementary Data 1. *S. youssoufiensis* OUC6819 (former name: *Streptomyces* sp. CHQ-64) was isolated from reed rhizosphere soil collected from the mangrove conservation area of Guangdong province, China[22]. *Streptomyces* sp. NRRL F-5123 and *S. aidingensis* CGMCC 4.5739 were obtained from Agricultural Research Service Culture Collection (NRRL) and China General Microbiological Collection Center (CGMCC), respectively. The standards of cyclodipeptides were purchased from GL Biochem Ltd (Shanghai, China). Drimentine A, C and G were provided by Prof. Ang Li (Shanghai Institute of Organic Chemist Chemistry, Chinese Academy of Sciences). The prenyl donor farnesyl diphosphate (FPP) was purchased from Sigma-Aldrich. Common biochemicals and chemicals were purchased from standard sources for laboratory use.

All DNA manipulations were performed according to standard procedures[36] or manufacturer's instruction. Plasmid extractions and DNA purification were carried out using commercial kits (OMEGA, BIO-TEK). Both primer synthesis and DNA sequencing were performed at TSINGKE Biotech Co. Ltd. (Qingdao, China). HR-ESIMS was carried out on Thermo LTQ Orbitrap XL mass spectrometer. NMR data was recorded with Agilent-DD2-500 or Bruker Avance III 600 spectrometers.

**Bioinformatic analysis**. ORF assignments and their proposed function were accomplished by using FramePlot4.0 beta[37] (http://nocardia.nih.go.jp/fp4). Sequence comparisons and database searches were accomplished with BLAST programs[38] (http://blast.ncbi.nlm.nih.gov/Blast.cgi). The structural model of DmtB1 was generated by using I-TASSER server[24] (https://zhanglab.ccmb.med.umich.edu/I-TASSER).

**Expression of *dmt1–3* in *E.coli* and *Streptomyces* strains**. For the expression of *dmtB1–3* in *E. coli*, the responding genes were amplified by polymerase chain reaction (PCR) using primer pairs listed in Supplementary Table 1 from the genomes of *S. youssoufiensis* OUC6819, *Streptomyces* sp. NRRL F-5123 and *S. aidingensis* CGMCC 4.5739, respectively. The amplification products were digested with *Nde*I and *Xho*I, ligated into pET-28a resulting in pWLI611-613, which were transformed into *E. coli* BL21 (DE3) after confirmation by sequencing. For expression of *dmtC1* in *E. coli*, it was amplified from the *S. youssoufiensis* OUC6819 genome using the primer pair of DmtC1-FP/RP (Supplementary Table 1). The resulting PCR products were digested with *Eco*RI and *Xho*I, and ligated into pET-32a to give pWLI627, followed by sequencing and transforming into *E. coli* BL21 (DE3).

For the expressions in *Streptomyces* strains, the corresponding gene(s) was/were put under the control of the constitutive promoter $P_{gapdh}$ from *S. youssoufiensis* OUC6819. Construction was performed as follows: taking *dmtB1C1* expression plasmid as an example, $P_{gapdh}$ was amplified using primer pair of pGFP/pGRP (Supplementary Table 2) and was digested with *NsiI*; the *dmtB1C1* fragment was amplified with the primer pair of dmtB1C1-FP/dmtC1-RP (Supplementary Table 2) and digested with *XbaI*; and then these two digested fragments were ligated and cloned into the *NsiI* and *XbaI* sites of pSET152C to give pWLI615. Similarly, the fragments of *dmtA1B1C1*, *dmtB2C2*, *dmtA2B2C2*, *dmtB3C3*, and *dmtA3B3C3* were cloned into the same sites of pSET152C to obtain pWLI616-620 (Supplementary Data 1), respectively. After confirmation by sequencing, the resulting plasmids were passed through *E. coli* ET12567/pUZ8002 and then introduced into *S. coelicolor* M1146 via conjugation[39].

**Construction of site-directed mutagenesis variants.** Site-directed mutagenesis was accomplished by using SOE-PCR method[40]. All PCR primers are listed in Supplementary Table 3 (for DmtB1 and DmtB2 variants construction) and Supplementary Table 4 (for DmtA1 variants construction). Taking the variant L185F as an example, the recombinant plasmid pWLI611 was used as template in the first and second PCR reactions, using the primer pairs of DmtB1-FP/DmtB1L185F-RP and DmtB1-RP/DmtB1L185F-FP. The resulting PCR products were purified and served as a template for the third round of PCR using the primer pair of DmtB1-FP/DmtB1-RP. After digestion with *NdeI* and *XhoI*, the amplified products were ligated into pET-28a and transformed into *E. coli* DH5α. The final recombinant plasmid was sequenced to ensure the accuracy of the intended gene mutation and then transformed into *E. coli* BL21 (DE3). For the purpose of probing the key residues in DmtA1, the resulting variants were introduced into Δ*dmtA1* via intergenic conjugation.

**Identification of DKP compounds formed by DmtB1–3.** Overnight cultures of *E. coli* BL21 (DE3) cells containing expression constructs of DmtB1–3 were used to inoculate [1:50 (v/v)] into 50 mL of M9 minimal medium[26] containing 50 μg mL$^{-1}$ kanamycin. Expression of the recombinant proteins was induced at an OD$_{600}$ of approximately 0.8 by addition of isopropyl-β-D-1-thiogalactopyranoside (IPTG, 0.1 mM final concentration), and cultivation was continued for additional 20 h at 16 °C. The cells were collected by centrifuging at 10,000×*g* for 20 min, resuspended in 2 mL of ice-cold binding buffer (0.05 M Tris-HCl, 0.5 M NaCl, pH 7.5, containing cOmplete™ protease inhibitor cocktail) and lysed by sonication at the same time. Proteins in crude extracts were detected by western blotting according to manufacturer's instruction. The His$_6$-tagged proteins were detected with primary antibody (Cell Signaling Technology: CST, # 2365 at 1:1000) and visualized using SuperSignal™ West Femto Substrate Trial Kit (Thermo Scientific, # 34094).

The supernatants of the cultures were extracted three times with an equal volume of ethyl acetate. The combined extracts were concentrated in vacuo and dissolved in 1 mL of MeOH. HPLC analyses were conducted using an Agilent 1260 HPLC system equipped with a YMC-Pack ODS-AQ C18 column (150 mm × 4.6 mm, particle size of 5 μm, pore size of 120 Å) under the following program: solvent A consisting of 0.1% (v/v) formic acid and ddH$_2$O, solvent B consisting of 0.1% (v/v) formic acid and acetonitrile; 10% B (0–5 min), 10% to 50% B (5–25 min), 100% B (25–35 min), at a flow rate of 1 mL min$^{-1}$ and UV detection at 280 nm. DKPs were quantified on the basis of their peak area at 280 nm using calibration curves obtained with authentic standards.

**Genomic library construction and library screening.** Genomic DNA of *S. youssoufiensis* OUC6819 was partially digested with *Sau*3AI, and fragments with the size of 40–50 kb were recovered and dephosphorylated with FastAP (Thermo Scientific, Pittsburgh, USA), and then ligated into SuperCos1 that was pretreated with *XbaI*, dephosphorylated, and digested with *Bam*HI. The ligation product was packaged into lambda particles with the MaxPlax Lambda Packaging Extract (Epicenter, Madison, WI, USA) as per the manufacture's instruction and plated on *E. coli* Top10. The titer of the primary library was about 5 × 10$^6$ cfu per μg of DNA. The primer pair used for cosmid library screening is listed in Supplementary Table 5.

**Gene manipulation.** Gene inactivation in *S. youssoufiensis* OUC6819 was performed using the REDIRECT Technology according to the literature protocol[41]. The *aac(3)IV-oriT* resistance cassette was amplified with appropriate primers (Supplementary Table 6) using pIJ773 as template and was transformed into *E. coli* BW25113/pIJ790 containing cosmid pWLI614 to replace an internal region of *dmtA1*, *dmtB1* and *dmtC1*, respectively, resulting in mutant cosmids pWLI621-623, respectively. pWLI621-623 were passed through *E. coli* ET12567/pUZ8002 and then introduced into *S. youssoufiensis* OUC6819 by intergenic conjugation using mycelia as recipients. A 24-h-old 5 mL of *S. youssoufiensis* OUC6819 culture in RARE3 medium (10 g L$^{-1}$ glucose, 4 g L$^{-1}$ yeast extract, 10 g L$^{-1}$ malt extract, 2 g L$^{-1}$ Bacto peptone, 2 g L$^{-1}$ MgCl$_2$·H$_2$O, 5 g L$^{-1}$ glycerol, 5 g L$^{-1}$ glycine) was diluted 1:10 followed by additional incubation at 30 °C for 24 h. The mycelia were then collected by centrifugation and resuspended in equal volume of RARE3 medium, and were homogenized and fragmented by sonication with an ultrasonic processors VCX750 (Sonics and Materials Inc, PA, USA). After incubation at 30 °C

for another 18 h, the mycelia were treated again as described above, resulting in recipient cells. The donor cells, *E. coli* ET12567/pUZ8002 carrying the desired conjugative plasmid, and the recipient cells were mixed in 1:10 ratio and plated on ISP4[39] agar supplemented with 20 mM MgCl$_2$. The desired mutants were selected by the apramycin-resistant and kanamycin-sensitive phenotype, and were further confirmed by PCR (Supplementary Table 6, Supplementary Figs. 20–22). For genetic complementation and overexpression experiments, the *dmtA1*, *dmtB1*, and *dmtC1* expression plasmids were constructed in a similar way as that for the *dmtB1C1* expression plasmid, generating pWLI624-626, respectively. pWLI624-626 were passed through *E. coli* ET12567/pUZ8002 and then introduced into the wild-type and mutant strains for gene overexpression and genetic complementation, respectively, via intergenic conjugation as described above.

**Production and analysis of fermentation products.** *Streptomyces* strains, including heterologous expression strains, mutant strains, genetic complementary strains, and overexpression strains, were incubated on a rotatory shaker at 30 °C in 250 mL Erlenmeyer flasks each containing 50 mL of fermentation medium composed of 1% soluble starch, 2% glucose, 4% corn syrup, 1% yeast extract, 0.3% beef extract, 0.05% MgSO$_4$·7H$_2$O, 0.05% KH$_2$PO$_4$, 0.2% CaCO$_3$, and 3% sea salt, pH 7.0. After 7 days of cultivation, the fermentation cultures were harvested by centrifugation. The supernatants were extracted twice with an equal volume of ethyl acetate, and the combined EtOAc extracts were concentrated in vacuo to afford residue A. The precipitated mycelia were extracted twice with acetone. The extracts were combined, and acetone was evaporated in vacuo to yield residue B. The combined residues were dissolved in methanol and filtered through a 0.2 μm filter. The resulting fermentation products were detected by HPLC analysis, using a YMC-Pack ODS-AQ C18 column with UV detection at 300 nm under the following program: solvent (phase A, 0.1% formic acid in ddH$_2$O; phase B, 100% acetonitrile supplemented with 0.1% formic acid); 20% B (0–5 min), 20% to 100% B (5–45 min), 100% B (45–55 min), at a flow rate of 1 mL min$^{-1}$.

**Isolation of pre-drimentines and drimentines.** The Δ*dmtA1* mutant strain and *S. coelicolor* M1146 expressing *dmtA1B1C1* or *dmtA2B2C2* were fermented in volume of 20 L to isolate compounds **6–8** and **13** (from Δ*dmtA1*), **9** (from M1146/pWLI616), **10** and **11** (from M1146/pWLI618), respectively. The fermentation cultures were treated as described above. The residues were applied to reversed-phase C18 open column, eluting with a gradient eluent of 20%–100% methanol to give five fractions (Fr.1~Fr.5) for each fermentation culture. Compounds **6–8** and **13** were obtained by further separation of the Fr.4 from Δ*dmtA1*, eluting with linear gradient from 80 to 100% B/A (phase A: ddH$_2$O; phase B: acetonitrile, 2 mL min$^{-1}$, UV detection at 300 nm) in 40 min using a semi-preparative HPLC column (YMC-Pack ODS-AA C18 column, 120 Å, 250 × 10 mm, 5 μm). Compounds **9–11** were also obtained by separation of the Fr.4 from M1146/pWLI616 (for **9**) and M1146/pWLI618 (for **10–11**) under the same HPLC conditions. The structures of the compounds were characterized by HR-ESIMS and NMR spectroscopy. All NMR spectra were processed with MestReNova.6.1.0 (Metrelab), and chemical shifts were referenced to those of the solvent DMSO-$d_6$ signals.

**Protein expression and purification.** 10 mL overnight culture of *E. coli* BL21 (DE3) harboring DmtB1 or DmtC1 expression plasmid was inoculated into 1 L of LB medium (containing 50 μg mL$^{-1}$ kanamycin or ampicillin) and grown at 37 °C with shaking. Expression was induced at an OD$_{600}$ of approximately 0.6 by addition of IPTG (0.1 and 0.05 mM final concentration for DmtB1 and DmtC1, respectively), and cultivation was continued for additional 16 hrs at 16 °C.

The cells were pelleted by centrifugation (15 min at 8000×*g*) and resuspended in 30 mL of binding buffer A (0.15 M NaCl, 5.0% glycerol (v/v), 1.0% (v/v) Triton X-100, 0.1 M Tris-HCl, pH 8.0, containing cOmplete™ protease inhibitor cocktail) for DmtB1 expression and (0.05 M Tris-HCl, 0.5 M NaCl, pH 7.5, containing cOmplete™ protease inhibitor cocktail) for DmtC1 expression. The resuspended cells were lysed by sonication in an ice-water bath with an ultrasonic processors VCX750 (Sonics and Materials Inc, PA, USA), and centrifuged at 10,000×*g* for 30 min at 4 °C. The supernatant was applied to a HisTrap HP column (1 mL, GE Healthcare) and the His-tagged DmtC1 protein was eluted with a linear gradient of imidazole (30–500 mM) in the binding buffer using an ÄKTA Purifier system. After SDS–PAGE analysis, fractions containing pure DmtC1 were pooled, concentrated and exchanged to Tris buffer (0.025 M Tris-HCl, 0.02 M NaCl, and 10.0% glycerol, pH 7.5) by using Amicon Ultra-15 30-kDa cutoff centrifugal concentrator (Millipore). And in case of DmtB1 and its variants, proteins were treated with RNase A and further subjected to a Heparin HP column (1 mL, GE Healthcare). Proteins were eluted with a linear gradient from 100% buffer B (0.1 M Tris-HCl pH 8.0, 0.05 M NaCl) to 100% buffer C (0.1 M Tris-HCl, pH 8.0, 1.0 M NaCl) over 30 min at a flow rate of 0.5 mL min$^{-1}$. Fractions containing pure proteins, as evidenced by SDS-PAGE, were concentrated and the buffer was exchanged to buffer D (0.1 M Tris-HCl, pH 8.0, 0.15 M NaCl, 5.0% glycerol (v/v) and 10 mM beta-mercaptoethanol) using a Amicon Ultra-15 10-kDa cutoff centrifugal concentrator (Millipore).

**assays.** A typical assay for DmtB1 and its variants was completed using established methods[18]. Briefly, recombinant DmtB1 or its variants (15 μM) was incubated with

50 μM tRNA mix from *E.coli* (Sigma, # R1753), aminoacyl-tRNA synthetase from *E.coli* (250 U, Sigma, # A3646), 5 mM amino acids (L-Trp, L-Val, L-Pro, L-Leu, and L-Ile), 5 mM ATP, 10 mM $MgCl_2$, 30 mM KCl, 100 mM Tris-HCl buffer (pH 8.0) and RNase inhibitor (20 U, Roche) were incubated in a total volume of 100 μL for 20 h at 30 °C. Reactions were quenched with TFA (2 μL, 2% v/v), centrifuged ($17,000 \times g$) for 30 min and the supernatants were subjected to LC-MS². Assays were conducted in duplicates.

The Agilent series 1290 HPLC system equipped with Agilent 6430 triple quadrupole mass spectrometry was used for the LC-MS² analysis. All separations were carried out on a reversed-phase Thermos Hypersil GOLD C18 column (100 mm × 2.1 mm, 1.9 μm, 175 Å, Thermo Scientific Inc., USA) under the following program: solvent A consisting of 0.1% (v/v) formic acid and $ddH_2O$, solvent B consisting of 0.1% (v/v) formic acid and acetonitrile; 10% B (0–5 min), 10% to 50% B (5–30 min), 100% B (30–40 min), at a flow rate of 0.2 mL min⁻¹. The optimized mass spectrometric parameters including fragmentor voltage (FG), collision energy (CE), and multiple reaction monitoring (MRM) transitions for each compound are shown in Supplementary Table 8. Agilent Mass Hunter software rev. B.03.01 was used for analysis.

The enzymatic assay of DmtC1 was carried out in 50 mM Tris-HCl buffer (pH 8.0) with 2.5 mM $MgCl_2$, containing 10 μM DmtC1, 1 mM DKP, and 0.2 mM FPP. The optimal assay conditions were obtained at 30 °C. After 1 hr, the reaction was quenched by the addition of equal volume of methanol and mixed by vortexing. The mixture was centrifuged at $17,000 \times g$ for 20 min to remove protein. The supernatant was then subjected to HPLC analysis under the identical conditions used for analyzing the fermentation products of the *Streptomyces* strains.

## Data availability

The nucleotide sequences of the *dmt1* locus, its upstream gene *orf(-1)* and downstream genes *orf1* and *orf2* have been deposited in the GenBank database under accession numbers of MG776357 [https://www.ncbi.nlm.nih.gov/nuccore/MG776357], MH666061 [https://www.ncbi.nlm.nih.gov/nuccore/MH666061], MH666062 [https://www.ncbi.nlm.nih.gov/nuccore/MH666062] and MH666063 [https://www.ncbi.nlm.nih.gov/nuccore/MH666063], respectively. All relevant data supporting the findings of this study are available within the paper and its supplementary information files. Additional data are upon reasonable request.

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

## Acknowledgements

We would like to thank Prof. Christopher Corre from the University of Warwick
and Prof. Liangcheng Du from the University of Nebraska-Lincoln for
constructive suggestions and discussions, Prof. Ang Li from Shanghai Institute of
Organic Chemist Chemistry, CAS, for kindly providing us DMT standards, Prof.
Mervyn J. Bibb (John Innes Centre, UK) for kindly providing us *S. coelicolor* M1146,
and the ARS Culture Collection (NRRL) for kindly providing *Streptomyces* sp. NRRL
F-5123. This work was supported by grants from the National Natural Science
Foundation of China (31171201, 31570032, 31711530219, 41506157, and
81561148012), and the NSFC-Shandong Joint Foundation (U1706206 and
U1406403).

## Author contributions

T.Y. and J.L. performed all the experiments. Z.L. and T.L. assisted in genetic manip-
ulation of *S. youssoufiensis* OUC6819 and bioinformatics analysis. H.L. was involved in
NMR analysis. Q.C., T.Z., D.L., and Q.G. isolated the *S. youssoufiensis* OUC6819 strain
and identified DMTG and F.T.Y. was involved in the draft manuscript writing. WL
supervised the whole work and wrote the manuscript. All authors read and approved the
final manuscript.

## Additional information

018-06411-x.

**Competing interests:** The authors declare no competing interests.

**Reprints and permission** information is available online at http://npg.nature.com/
reprintsandpermissions/

**Publisher's note:** Springer Nature remains neutral with regard to jurisdictional claims in
published maps and institutional affiliations.

