## [Peer Review File · Nature Communications]

Reviewers' comments:

Reviewer #1 (Remarks to the Author):

The paper presents the cyclodipeptide synthase-dependent biosynthesis of terpinylated diketopiperazines named drimentines (DMTs). The authors identified in *Streptomyces youssoufiensis* a three-gene cluster, *dtm1*, which is homologous to two clusters, *dtm2* and *dtm3*, respectively found in *Streptomyces* sp. NRRL F-5123 and *Streptomyces aidingensis* GCMCC 4.5739 (the last two genomes are available in GenBank). The three homologous genes encode a cyclodipeptide synthase (DmtB), a prenyltransferase (DmtA) and a terpene cyclase (DmtC), respectively. The authors convincingly show that DmtB catalyses the formation of cyclo(L-Trp-L-Xaa), DmtC adds a farnesyl diphosphate group on the cyclodipeptides and DmtA acts as a terpene cyclase.

The results are significant, the arguments are sound and the conclusions are supported by appropriate experiments. However, they must revise their manuscript to keep a main message and highlight the originality of their work, namely the characterization of two original cyclodipeptide-modifying enzymes.

The paragraph "Genome mining" should be shortened and modified taking into account the paper Gondry et al (2018) *Front Microbiol.* 2018 Feb 12;9:46. Indeed, in this paper, the authors showed that several CDPSs catalyse cyclo(L-Trp-L-Xaa) and DmtB2 was already characterized as a cyclo(L-Trp-L-Pro)-synthesizing enzyme.

The paragraph "Probing the molecular...mutagenesis" should be significantly shortened because it is not completely relevant with the main message. The work is significant, the results could be detailed in the SI.

Concerning the following paragraphs, they are convincing and original. The work in *Streptomyces* strains seems very well conducted. Could the authors specify how all the compounds in Figures 5 and 6 were identified? Not only, HPLC analyses with retention times? The results of DmtAs variants are interesting to detail because they were constructed to probe the enzyme mechanism and not the enzyme specificity (as done for CDPS variants).

Concerning the conclusion on methylation steps, the authors should comment on genes present upstream and downstream of the studied three-gene operons. Regarding *dtm2*, one gene encodes a putative methyltransferase. In addition, others genes encodes putative cyclodipeptide oxidase and so on... The description of the clusters must be completed.

Reviewer #2 (Remarks to the Author):

In this work Yao et al. characterize three cyclodipeptide synthase (CDPS) enzymes, establishing their DKP biosynthetic products and using site-directed mutagenesis experiments to propose residues involved in substrate recognition. They also characterize other genes clustered with these CDPSs, including genes encoding prenyltransferases and cyclases that act to tailor DKPs. Their topic is of interest to the broad biosynthesis and natural product community, yet aspects pertaining to the CDPS study are not of suitable novelty or rigor to warrant publication in *Nature Communications*. These issues related to novelty and rigor are commented upon below:

Novelty of work: Although the authors cite multiple 2018 papers (e.g. reference #18 + 19), they appear unaware of the recent work by Gondry and co-workers (*Frontiers in Microbiology* 2018, 9:1-14; PubMed: <https://www.ncbi.nlm.nih.gov/pubmed/29483897>). Multiple key CDPS findings from the Yao et al. manuscript are duplicative of this *Frontiers* paper. Specifically, the *Frontiers* paper established the biosynthetic function of the CDPS from *Streptomyces* sp. NRRL F-5123; the biosynthetic function of this same enzyme was also presented by Yao et al. in the current manuscript using the same methods. Additionally, work by Gondry and co-workers demonstrated production of novel cyclo(L-Trp-L-Xaa) molecules, including one of the molecules (cyclo-L-Trp-L-Ile) reported as novel by Yao et al. This finding by Gondry's group as well as other enzymes in their *Frontiers* paper negate the claim of Yao et al. that "all other characterized CDPSs synthesizing Trp-containing DKPs only produce cyclo(L-Trp-L-Trp) with high specificity" (page 13).

Rigor of work: The authors make a variety of conclusions about substrate recognition and specificity based on site-directed mutagenesis experiments for which only *in vivo* data are reported (pages 6-7). In many of these cases, their conclusions about impacts of mutagenesis are based on decreases/increases in production of DKP product by only 1.5-to-2 fold. With their *in vivo* system, production of DKPs by CDPSs are dependent not only upon the CDPS itself but also upon cellular availability of specific aa-tRNA substrates, whose titers could vary based upon a variety of feedback loops and other factors. These types of effects might account for the magnitude of decreases/increases in DKP production reported by the authors as significant findings. The rigor of these experiments would be substantially improved by adopting an *in vitro* approach toward probing substrate specificity, so that variables such as aa-tRNA substrate concentration could be more closely controlled. This *in vitro* approach has been the accepted standard methodology in earlier substrate specificity studies of CDPSs.

Thank you so much for your valuable comments regarding our manuscript entitled “Genome mining of cyclodipeptide synthases unraveling an unusual tRNA-dependent diketopiperazine-terpene biosynthetic machinery” (NCOMMS-18-05935A). We appreciate your warm work earnestly. All the comments are very helpful for revising and improving our manuscript. We have considered the comments very carefully and have made changes accordingly. The changes are highlighted in a yellow background. Point-by-point responses to the issues raised by the reviewers are listed below. We believe the revisions have resulted in an improved manuscript.

Reviewer #1 (Remarks to the Author):

The paper presents the cyclodipeptide synthase-dependent biosynthesis of terpinylated diketopiperazines named drimentines (DMTs). The authors identified in *Streptomyces youssoufiensis* a three-gene cluster, *dtm1*, which is homologous to two clusters, *dtm2* and *dtm3*, respectively found in *Streptomyces* sp. NRRL F-5123 and *Streptomyces aidingensis* GCMCC 4.5739 (the last two genomes are available in GenBank). The three homologous genes encode a cyclodipeptide synthase (DmtB), a prenyltransferase (DmtA) and a terpene cyclase (DmtC), respectively. The authors convincingly show that DmtB catalyses the formation of cyclo(L-Trp-L-Xaa), DmtC adds a farnesyl diphosphate group on the cyclodipeptides and DmtA acts as a terpene cyclase.

The results are significant, the arguments are sound and the conclusions are supported by appropriate experiments. However, they must revise their manuscript to keep a main message and highlight the originality of their work, namely the characterization of two original cyclodipeptide-modifying enzymes.

Response:

We would like to thank you for your constructive and valuable comments and suggestions! We have to the best of our abilities responded to them.

The paragraph “Genome mining” should be shortened and modified taking into account the paper Gondry et al (2018) Front Microbiol. 2018 Feb 12;9:46. Indeed, in this paper, the authors showed that several CDPSs catalyse cyclo(L-Trp-L-Xaa) and DmtB2 was already characterized as a cyclo(L-Trp-L-Pro)-synthesizing enzyme.

Response:

Thank you for your constructive suggestion! We have revised the introduction section with taking into account the paper by Gondry et al (Front Microbiol. 2018 Feb 12;9:46, which was published right before our submission) and by Liu et al (Appl. Microbiol. Biotechnol. 2018 Mar 24 102:4435–4444, which was published after our submission). The paragraph “Genome mining” has been shortened as suggested, and Figure 3 has been modified by moving the result of DmtB2 to the supporting information (Figure S2). We all found the

CDPS from Streptomyces sp. NRRL F-5123, named DmtB2 in our manuscript, synthesizes cWP as the major product in E. coli. The minor product(s) were reported to be cWF and cWG in Gondry's paper, and was found to be cWL in Liu's paper and in our experiment (Figure S2, panel ii).

The paragraph "Probing the molecular....mutagenesis" should be significantly shortened because it is not completely relevant with the main message. The work is significant, the results could be detailed in the SI.

Response:

Thank you for your suggestion! We have significantly shortened the paragraph "Probing the molecular....mutagenesis", and have removed the description for variants of F184A, A199S, A199N, V152M, N159A, E206A, E206P, H203A, K155A, K155E, K155L and K155R from the main text and have detailed them in the supporting information (Figure S6) as suggested. In addition, the in vivo mutagenesis results of DmtB2 (Figure 4c in the original manuscript) has been moved to Figure S7a, and the in vitro data of DmtB1, DmtB1-L185F and DmtB1-V205M has been added as Figure 4c in our revised manuscript (suggested by Reviewer 2).

Concerning the following paragraphs, they are convincing and original. The work in *Streptomyces* strains seems very well conducted. Could the authors specify how all the compounds in Figures 5 and 6 were identified? Not only, HPLC analyses with retention times?

Response:

*Thanks for your valuable comments and suggestion! As described in the paragraph "Isolation of pre-drimentines and drimentines from Δ dmtA1 and heterologous expression strains" in method section, compounds **6-8** and **13** were isolated from the fermentation broth of the Δ dmtA1 mutant strain and were identified via HRMS and 1D and 2D NMR analysis (Figure S16-S19; Table S11), and their production in *S. coelicolor* M1146 expressing dmtB1C1 (Figure 5 panel ii) were identified by comparison with those in the Δ dmtA1 mutant strain (Figure 6, panel ii). Compounds **9** and **12** were previously isolated and identified from the fermentation broths of the wild type strain *S. youssoufiensis* OUC6819 (Che, Q. et al. Org Lett 2012,14, 3438-3441); production of **9** in M1146 expressing dmtA1B1C1 (Figure 5 panel iii) was identified by isolation from the fermentation broths of M1146/pWLI616, followed by HRMS and 1H NMR (Figure S10); production of **12** in *S. youssoufiensis* OUC6819 was confirmed by HRMS (Figure S15). Compounds **10** and **11** were isolated from the fermentation broths of M1146/pWLI618 (Figure 5 panel v), and were identified by comparison with the authentic standards (Figure*

S9), as well as HRMS and ¹H NMR analysis (Figure S11 and S12). Thank you!

The results of DmtAs variants are interesting to detail because they were constructed to probe the enzyme mechanism and not the enzyme specificity (as done for CDPS variants). Concerning the conclusion on methylation steps, the authors should comment on genes present upstream and downstream of the studied three-gene operons. Regarding dmt2, one gene encodes a putative methyltransferase. In addition, others genes encodes putative cyclodipeptide oxidase and so on... The description of the clusters must be completed.

Response:

Thanks for your valuable comments and suggestion! We have completed the description about the surrounding genes of dmt1-3 in the paragraph of "Inspection of the genetic contexts of dmtB1-3 disclosing a novel CDPS-dependent pathway", and Figure S8 (Comparison of the surrounding genes of dmt1-3 and their predicted functions) has been added in the supporting information. The discussion has been revised accordingly.

Reviewer #2 (Remarks to the Author):

In this work Yao et al. characterize three cyclodipeptide synthase (CDPS) enzymes, establishing their DKP biosynthetic products and using site-directed mutagenesis experiments to propose residues involved in substrate recognition. They also characterize other genes clustered with these CDPSs, including genes encoding prenyltransferases and cyclases that act to tailor DKPs. Their topic is of interest to the broad biosynthesis and natural product community, yet aspects pertaining to the CDPS study are not of suitable novelty or rigor to warrant publication in Nature Communications.

Response:

We would like to thank you for your constructive comments and suggestions! We have to the best of our abilities responded to them. Thanks a lot!

These issues related to novelty and rigor are commented upon below:

Novelty of work: Although the authors cite multiple 2018 papers (e.g. reference #18 + 19), they appear unaware of the recent work by Gondry and co-workers (Frontiers in Microbiology 2018, 9:1-14; PubMed: <https://www.ncbi.nlm.nih.gov/pubmed/29483897>). Multiple key CDPS findings from the Yao et al. manuscript are duplicative of this Frontiers paper. Specifically, the Frontiers paper established the biosynthetic function of the CDPS from *Streptomyces* sp. NRRL F-5123; the biosynthetic function of this same enzyme was also presented by Yao et al. in the current manuscript using the same methods. Additionally, work by Gondry and co-workers demonstrated production of novel

cyclo(L-Trp-L-Xaa) molecules, including one of the molecules (cyclo-L-Trp-L-Ile) reported as novel by Yao et al. This finding by Gondry's group as well as other enzymes in their Frontiers paper negate the claim of Yao et al. that "all other characterized CDPSs synthesizing Trp-containing DKPs only produce cyclo(L-Trp-L-Trp) with high specificity" (page 13).

Response:

Thanks for your comments! We didn't realize the finding about the CDPS (dmtB2 in our manuscript) from Streptomyces sp. NRRL F-5123 by Gondry and co-workers as the paper was published right before our submission. Sorry about this! We have revised our manuscript with full consideration of the recently published work about dmtB2 (Gondry et al Front Microbiol. 2018 Feb 12;9:46, which was published right before our submission; Liu et al Appl. Microbiol. Biotechnol. 2018 Mar 24 102:4435–4444, which was published after our submission). Figure 3 has been modified by moving the result of DmtB2 to the supporting information (Figure S2). The paragraph "Genome mining" have been significantly shortened. The claim about novel cyclo(L-Trp-L-Xaa) group has been deleted. Noticeably, DmtB1 is the first cWX-synthesizing CDPS with cWV being the major product. Systematic mutagenesis experiments were carried out, and the important contribution of residues lining the substrate binding pockets P1 and P2 to the incorporation of the second aa-tRNA was established. More importantly, two original cyclodipeptide-modifying enzymes were characterized, unraveling a novel CDPS-dependent biosynthetic machinery.

Rigor of work: The authors make a variety of conclusions about substrate recognition and specificity based on site-directed mutagenesis experiments for which only in vivo data are reported (pages 6-7). In many of these cases, their conclusions about impacts of mutagenesis are based on decreases/increases in production of DKP product by only 1.5-to-2 fold. With their in vivo system, production of DKPs by CDPSs are dependent not only upon the CDPS itself but also upon cellular availability of specific aa-tRNA substrates, whose titers could vary based upon a variety of feedback loops and other factors. These types of effects might account for the magnitude of decreases/increases in DKP production reported by the authors as significant findings. The rigor of these experiments would be substantially improved by adopting an in vitro approach toward probing substrate specificity, so that variables such as aa-tRNA substrate concentration could be more closely controlled. This in vitro approach has been the accepted standard methodology in earlier substrate specificity studies of CDPSs.

Response:

Thanks so much your valuable suggestion! For the in vivo experiments, all of them were performed at least in triplicates under the same conditions (Figure 4b and 4d), and the

protein expression levels were also compared to that of the wild type enzyme by western blot analysis (Figure 4d). Therefore, the results were repeatable and comparable although the changed values may not be very significant. To give a clearer picture, the ratios of cWP (2) vs cWV (3) and cWL (4) vs cWV (3) have been calculated and has been added in supporting information (Table S9). The results showed, in comparison to that of the wild type DmtB1, the ratio of 2 vs 3 synthesized by DmtB1-L185F was changed from 1:13.1 to 22.8:1, and the ratio of 4 vs 3 synthesized by L185F and V205M was changed from 1:10 to 13:1 and 1.1:1, respectively. Therefore, the *in vivo* data clearly indicated L185 and V205 are probably important for substrate specificity of DmtB1.

We agree that production of DKPs by CDPSs *in vivo* depends on cellular availability of specific aa-tRNA substrates, whose titers could vary based upon a variety of feedback loops and other factors. So *in vitro* experiments of DmtB1 and its variants (L185F and V206M) have been carried out to more closely control the concentrations of aa-tRNA substrates as suggested. The result has been added as Figure 4c in our revised manuscript. It showed that the 2-forming ability of L185F was improved by 2.5-fold (panel ii), and no obvious change was observed for the 4-forming ability of V206M (panel iii) in comparison to the wild type DmtB1; simultaneously, the 3-forming ability of both variants was significantly decreased by about 43-fold (L185F, panel ii) and 6-fold (V205M, panel iii). As indicated in Table S9, the ratio of 2 vs 3 formed by L185F was changed from 1:518.8 to 1:4.3, and the ratio of 4 vs 3 by L185F and V205M was changed from 1:33.3 to 1:4.1 and 1:5, respectively. While the major product generated by the variants are different *in vivo* and *in vitro*, which was also observed for the study of AlbC (Gondry et al *Nucleic Acids Research*, 2014, 42(11): 7247–7258) and could be explained by the variable cellular availability of each aa-tRNA substrate, the changes in the ratios of the products are consistent. Therefore, the importance of L185 and V205 for substrate specificity can be safely established. Thanks again!

We hope the response above address the comment satisfactorily. We really appreciate all your nice help!

Best regards,

Yours sincerely,

Wenli LI

Reviewers' Comments:

Reviewer #1 (Remarks to the Author):

The authors responded satisfactorily to all my questions and comments.
The manuscript was amended accordingly.

Muriel Gondry

Reviewer #2 (Remarks to the Author):

The authors have adequately addressed concerns raised by the reviewers by conducting additional experiments and better taking into consideration pre-existing work in the field. The paper is now acceptable for publication.

REVIEWERS' COMMENTS:

Reviewer #1 (Remarks to the Author):

The authors responded satisfactorily to all my questions and comments.
The manuscript was amended accordingly.

Muriel Gondry

Response:

We would like to thank you for your supportive comments and recognition of our work!

Reviewer #2 (Remarks to the Author):

The authors have adequately addressed concerns raised by the reviewers by conducting additional experiments and better taking into consideration pre-existing work in the field. The paper is now acceptable for publication.

Response:

Thanks much for your supportive comments!